# Excessive Vitamin A Supplementation Increased the Incidence of Acute Respiratory Tract Infections: A Systematic Review and Meta-Analysis

**DOI:** 10.3390/nu13124251

**Published:** 2021-11-26

**Authors:** Yihan Zhang, Yifei Lu, Shaokang Wang, Ligang Yang, Hui Xia, Guiju Sun

**Affiliations:** Key Laboratory of Environmental Medicine and Engineering of Ministry of Education, Department of Nutrition and Food Hygiene, School of Public Health, Southeast University, Nanjing 210009, China; zhangyihan425@163.com (Y.Z.); luyifei3377@163.com (Y.L.); shaokangwang@seu.edu.cn (S.W.); yangligang2012@163.com (L.Y.); huixia@seu.edu.cn (H.X.)

**Keywords:** vitamin A supplementation, acute respiratory tract infections, meta-analysis, subgroups

## Abstract

(1) Background: vitamin A deficiency (VAD) is highly prevalent in children living in poor conditions. It has been suggested that vitamin A supplementation (VAS) may reduce the risk of acute respiratory tract infections (ARTI). Our study provides updates on the effects of oral VAS (alone) in children on ARTI and further explores the effect on interesting subgroups. (2) Methods: eight databases were systematically searched from their inception until 5 July 2021. The assessments of inclusion criteria, extraction of data, and data synthesis were carried out independently by two reviewers. (3) Results: a total of 26 randomized trials involving 50,944 participants fulfilled the inclusion criteria. There was no significant association of VAS with the incidence of ARTI compared with the placebo (RR 1.03, 95% CI 0.92 to 1.15). Subgroup analyses showed that VAS higher than WHO recommendations increased the incidence of ARTI by 13% (RR 1.13, 95% CI 1.07 to 1.20), and in the high-dose intervention group, the incidence rate among well-nourished children rose by 66% (RR 1.66, 95% CI 1.30 to 2.11). (4) Conclusions: no more beneficial effects were seen with VAS in children in the prevention or recovery of acute respiratory infections. Excessive VAS may increase the incidence of ARTI in children with normal nutritional status.

## 1. Introduction

Acute respiratory tract infection (ARTI), including acute upper respiratory infection (AURI), i.e., common cold, pharyngitis, and tonsillitis and acute lower respiratory infection (ALRI), i.e., pneumonia and bronchitis [1] are a major cause of morbidity and mortality worldwide [2]. In 2008, an estimated 8.795 million deaths occurred among children under the age of 5 in the world, of which 68% were caused by infectious diseases [3]. ARTI was responsible for an estimated 2.56 million deaths worldwide in 2017 and was the most common cause of death among children under 5 years old [4]. A systematic analysis estimated that in 2018 there were 10 million cases of influenza-virus-associated ALRI and around 35,000 total deaths from influenza-virus-associated ALRI, and most ALRIs occur in children and in people living in poor conditions [5]. The role of nutrition in supporting the immune system is well-established. Meanwhile, numerous mechanisms and clinical data suggest that vitamins, including vitamins A, B6, B12, C, D, and E, play important and complementary roles in supporting the immune system [6]. Inadequate intake of these nutrients is common, leading to a decline in resistance to infection, thereby increasing the burden of disease.

Vitamin A deficiency (VAD) is a condition resulting from inadequate absorption or ingestion of vitamin A, with a global burden estimated at 806,000 disability-adjusted life years (DALYs) [7]. Retinyl palmitate, the major dietary provitamin vitamin A, and β-carotene, the major dietary provitamin A carotenoids, are widely used to improve VAD [8]. Deficiency of this nutrient during pregnancy increases the risk of maternal night blindness and anemia and may be a cause of congenital malformations, childhood VAD can also cause xerophthalmia, lower resistance to infection and increase the risk of mortality [9]. WHO recommends vitamin A supplementation (VAS) with a dose of 100,000 IU (30 mg retinol activity equivalent, RAE) in infants aged 6–11 months and 200,000 IU (60 mg RAE) every 4–6 months for children aged 12–59 months living in settings where VAD is a public health problem [10].

With an interest in the intervention potential for vitamin-A-deficient populations, a number of randomized trials have tested the effectiveness of high-dose VAS for children aged from 6 months to 5 years. A meta-analysis by Brown and colleagues showed no evidence of a benefit from VAS on ALRI recovery in developing countries [11]. Another meta-analysis in 2003 showed that VAS was correlated with a slight but significant increase in respiratory tract infections (RTI) (RR 1.08, 95% CI: 1.05–1.11) [12]. However, a recent randomized control study demonstrated that VAS decreased the incidence rate of respiratory-related illnesses (*p* < 0.05) [13]. Swami and colleagues also observed a decline in morbidity due to ARTI after the mass distribution of vitamin A in first month (*p* < 0.001) [14]. In addition, high-dose VAS was found to have the opposite effect in children with different nutritional status—the incidence of ALRI was lower in underweight supplement-treated children but higher in the normal-weight group compared with a placebo [15]. No clear conclusion can be drawn as to the effect of VAS in the amelioration of ARTI, and the significance of the dose has not been considered; thus, in this study we aimed to include all randomized control studies related to this issue and tried to clarify whether vitamin A is effective in relation to ARTI or not.

## 2. Materials and Methods

The Preferred Reporting Items for Systematic Reviews and Meta-Analysis (PRISMA) guidelines were used to conduct this review [16].

### 2.1. Search Strategy and Study Selection

We conducted an electronic search of Pubmed, Web of science, EMBASE, Medline, the Cochrane Library, Science direct, Scopus databases, and Ovid from their inception up to 5 July 2021. We searched articles using the following terms: (Vitamin A or retinol or carotene) and (respiratory tract infection or acute lower respiratory infection or acute respiratory tract infection or acute upper respiratory infection or pneumonia or tonsillitis or Sore throat or pharyngitis or cold or bronchitis) and (Supplement or fortified or intervention) and (trial or double blind or single blind or controlled study or comparative study). The detailed search strings are shown in Appendix A.

### 2.2. Inclusion/Exclusion Criteria 

All studies were identified from databases and imported into EndNote X9 and repeated studies were deleted. Abstracts/titles/full texts were screened by two reviewers, Yihan Zhang and Yifei Lu, according to the inclusion and exclusion criteria.

The inclusion criteria: (1) randomized controlled trials (RCTs); (2) full text available; (3) reported on the association between VAS and ARTI; (4) reported on children aged 0 months to eleven years.

The exclusion criteria: (1) studies investigating the link between VAS and tuberculosis or chronic lung conditions such as chronic obstructive pulmonary disease and asthma; (2) studies evaluating the effects of food fortification, the consumption of foods rich in vitamin A, beta-carotene supplementation, or co-interventions (for example, multiple vitamin or mineral supplementation); (3) severe malnutrition; (4) signs of VAD, including xerophthalmia; (5) concurrent serious illness; (6) concurrent measles infection.

### 2.3. Definition of Outcomes

Primary outcomes: the primary outcome was the incidence of ARTI, including AURI, ALRI, and acute respiratory tract infection of unspecified location.

Secondary outcomes: The secondary outcomes included (1) the severity of ARTI—the mean number of episodes, the duration of the illness; (2) the incidence of ALRI; and (3) the severity of ALRI—hospitalization, days to resolution of symptoms.

### 2.4. Data Extraction and Quality Assessment

For this review, two investigators (Yihan Zhang and Ligang Yang) independently used a data collection sheet to extracted the required information from eligible studies, including the full name of the first author, year of study, location, study design, characteristics of participants (number of each group, age, sex), types of interventions (duration, dose, frequency), and outcome characteristics.

We used the Cochrane Collaboration’s risk-of-bias tool to assess the quality of the evidence, obtained seven categories, namely, random sequence generation, allocation concealment, blinding of participants and personnel, blinding of outcome assessment, incomplete outcome data, selective reporting, and other bias.

### 2.5. Data Synthesis and Analysis

STATA 11.0 software (Stata, College Station, TX, USA) was used for the meta-analysis. The severity of ARTI and ALRI was calculated based on the mean difference (MD) and their associated 95% confidence intervals (CI). When only the standard errors of the means (SEMs) were reported, we calculated SDs by multiplying SEMs by the square root of the sample size. When reporting the median and range we used the formula given in the Cochrane guidelines to obtain means and SDs [17]. For dichotomous outcomes, rate ratios (events per child-year) and risk ratios (events per child) used the same scale and could be interpreted in the same way; thus, we combined the rate ratios and risk ratios to calculate the incidence of ARTI and ALRI. Heterogeneity was assessed using the I2 statistic. When I2 < 50% and *p* < 0.1, we used fixed-effects models to pool outcomes, otherwise we used random-effects models (I2 ≥ 50%).

Subgroup analysis was performed to calculate the effect of dose (standard (up to 100,000 IU for children aged 0 to 11 months, and 200,000 IU for children aged 12 months to 11 years every 4–6 months) or high (greater than standard)), frequency (low (one dose 4-plus-month), medium (one dose every 4 months), or high (doses more than once in 4 months)), area, and three kinds of nutritional status on outcome measures: stunted, wasted, and normal. Variations among subgroups were evaluated on the basis of the *p*-value at 95% CI.

To assess the impact of each individual study on the overall estimates for the rest of the studies, the leave-one-out sensitivity analysis was repeated by deleting one study at a time to confirm that the findings were not affected by any individual study. We used Egger’s test to test for publication bias.

## 3. Results

### 3.1. Baseline Characteristics of Included Studies

Figure 1 illustrates the selection process of this study. Databases searches yielded 2309 records after duplicates were removed. We identified 141 relevant studies, searching by title and abstract. Twenty-six of the 141 original studies screened met the inclusion criteria for quantitative synthesis (meta-analysis). Of the 26 studies included, 10 studies reported the association between vitamin A and LRTI [11,18,19,20,21,22,23,24,25,26], 16 reported the association between vitamin A and ARTI [13,14,15,27,28,29,30,31,32,33,34,35,36,37,38,39]. Table 1 summarizes the characteristics of the included studies. We identified a total of 50,944 participants, 25,981 individuals on intervention, and 24,963 on placebo. The ages of the enrolled children ranged from 0 months to 11 years. The total dosage of vitamin A varied from 50,000 to 618,000 IU (15–185.4 mg RAE) for infants and from 100,000 to 1236,000 IU (30–370.8 mg RAE) for 1–11-year-olds. The baseline serum retinol levels of all participants were within the normal range and showed no significant difference between the intervention group and control group. The quality of each trial included in the study is listed in Table 2.

### 3.2. Association between Vitamin A and ARTI

#### 3.2.1. Incidence of ARTI

We found that seven studies showed no significant difference between the vitamin A group and the placebo group (RR 1.03, 95% CI 0.92 to 1.15; I2 = 74.4%, *p* = 0.001; Figure 2; [13,15,30,33,34,38,39]).

#### 3.2.2. The Severity of ARTI

Six studies shared the continuous outcome episodes of all symptoms and other continuous outcomes, including cough days, mean duration per episode, and mean duration per child, reported by three studies, respectively. Vitamin A intervention did not significantly change the mean number of episodes (WMD −0.35, 95% CI −1.25 to 0.55; I2 = 91.3%, *p* < 0.001; Figure 3a; [14,28,35,36,37,39]), mean cough days (WMD 0.12, 95% CI −1.43 to 1.67; I2 = 0.0%, *p* = 0.571; Figure 3b; [13,35,39]), mean duration per episode (WMD −0.07, 95% CI −0.50 to 0.37; I2 = 0.0%, *p* = 0.552; Figure 3c; [28,30,36]), or mean duration per child (WMD −5.19, 95% CI −12.81 to 2.42; I2 = 80.4%, *p* = 0.006, Figure 3d; [28,32,37]).

### 3.3. Association between Vitamin A and LRTI

#### 3.3.1. Incidence of LRTI

Six studies analyzed the incidence of LRTI. Meta-analysis did not show significant differences between the intervention group and the placebo group (RR 1.02, 95% CI 0.94 to 1.12; I2 = 35.1%, *p* = 0.173; Figure 4; [15,26,27,29,30,31]).

#### 3.3.2. The Severity of LRTI

Seven studies shared the outcome days to resolution of fever and days for respiratory rate to settle, and six studies reported on the days to normalization of oxygen saturation (SpO2) and days in hospital. There was no evidence of heterogeneity between studies for days to resolution of fever (I2 = 0.0%, *p* = 0.695), days for respiratory rate to settle (I2 = 0.0%, *p* = 0.568), days to normalization of SpO2 (I2 = 0.0%, *p* = 0.811) or days in hospital (I2 = 0.0%, *p* = 0.889), so a fixed-effect model was used. Supplementation with vitamin A did not significantly shorten days to resolution of fever (WMD −0.07, 95% CI −0.17 to 0.03; Figure 5a; [11,18,19,21,22,23,24]), days for respiratory rate to settle (WMD 0.00, 95% CI −0.17 to 0.17; Figure 5b; [11,18,19,21,22,23,24]), days to normalization of SpO2 (WMD 0.00, 95% CI −0.01 to 0.01; Figure 5c; [18,19,20,21,22,23]), or days in hospital (WMD 0.11, 95% CI −0.33 to 0.54; Figure 5d; [18,19,20,21,24,25]).

### 3.4. Subgroup Analysis

We found evidence of significant heterogeneity in outcome 1 and outcome 2; thus, we performed subgroup analysis to explore potential sources of heterogeneity. For the incidence of ARTI, subgroup analysis was carried out in terms of dose, frequency, area, and nutritional status (Table 3). High-dose (greater than standard) vitamin A intervention was associated with a 13% increase in the incidence of ARTI (RR 1.13, 95% CI 1.07 to 1.20) and standard dose did not show any significant difference (RR 0.82, 95% CI 0.64 to 1.03). Only medium-frequency (one dose every 4 months) intervention studies reported a combined 14% increase in ARTI incidence (RR 1.14, 95% CI 1.07 to 1.23). Three RCTs reported on the incidence of ARTI after VAS in groups based on nutritional status, in which the doses exceeded the WHO recommendations. It was observed that participants with normal nutritional status showed a higher risk of the incidence of ARTI after vitamin A intervention (RR 1.66, 95% CI 1.30 to 2.11), but those with a stunted and wasted nutritional status showed no significant difference between the vitamin A group and the placebo group (RR 0.82, 95% CI 0.46 to 1.47) (RR 0.59, 95% CI 0.21 to 1.67). No subgroup differences were observed between the two groups in terms of area. Dose, frequency, and area were considered as the stratified factors of outcome 2 (Table 4). 

### 3.5. Sensitivity Analysis

We conducted sensitivity analysis to investigate whether any single study significantly affected the pooled results and the results showed the stability of the meta-analysis (Figure 6 and Figure 7).

### 3.6. Publication Bias

For the outcomes of more than seven studies, we used the Egger test to test for publication bias. The Egger test showed no evidence of significant publication bias (*p* = 0.48, *p* = 0.46, *p* = 0.07).

## 4. Discussion

To the best of our knowledge, this is the first meta-analysis and systematic review that has included all randomized control studies (RCTs) on this issue, reporting on the effects of vitamin A on ARTI. Overall, our meta-analysis showed no evidence that VAS alters the incidence or course of ARTI, which was consistent with the results of previous meta-analyses which studied LRTI [40,41]. Stratified analyses were conducted to further explore the effect of vitamin A on interesting subgroups. We found that VAS exceeding the WHO’s recommended dose led to an increase in ARTI incidence, especially in children with good nutrition.

Vitamin A is an essential nutrient and it cannot be synthesized by the human body and must be obtained from dietary sources [42]. Oral VAS and food fortification are the most direct ways to provide vitamin A for people with vitamin A deficiency and the absorbed vitamin A is mainly stored in the liver [8]. In populations whose vitamin A availability from food is low, infectious diseases can precipitate VAD by decreasing intake, decreasing absorption, and increasing excretion [43]. Vitamin A has been described as an anti-infectious vitamin because of its role in regulating the human immune system and immune functions [44]. By improving immune function, vitamin A reduces mortality associated with measles, diarrhea, and other illnesses [45]. Studies in animals have revealed that VAD could lead to immunoglobulin dysregulation, squamous cell metaplasia, infectious disease, and death [46]. However, excessive intake of vitamin A can also lead to acute and chronic toxicity. Recently, several human studies have suggested an association between an excessive intake of vitamin A and increased bone fragility that potentially leads to osteoporosis [47,48]. Children are particularly sensitive to vitamin A, with daily intakes of 1500 IU/kg body weight reportedly leading to toxicity [49]. Borel and his colleagues also found that there is high individual variability in β-carotene bioavailability [50,51] and β -carotene is the most important dietary source of vitamin A and can be cleaved to form vitamin A after absorption in the intestine [52,53,54]. In the present meta-analysis, the dose of VAS exceeded the WHO’s recommended dose may lead to adverse effects, such as the 13% increase in ARTI incidence (*p* < 0.001, I2 = 19.3%). Furthermore, receiving vitamin A at the WHO-recommended dose might be effective in preventing ARTI, with a borderline *p*-value of 0.090. Of the 25 studies included, three studies [15,30,39] stratified participants by nutritional status and the intervention dose in all three studies exceeded the recommended dose. The pooled results showed that participants with normal nutritional status had a 66% increase in ARTI incidence (*p* < 0.001, I2 = 14.9%), but there was no evidence that VAS altered the ARTI incidence in stunted or wasted children. This indicates that high-dose vitamin A supplementation provided to children with adequate vitamin A stores might cause temporary immune dysregulation and lead to increase risks of infectious diseases.

VAS programs began in the 1990s in response to evidence demonstrating the association between VAD and increased childhood mortality [55,56]. At present, more than 80 countries worldwide are implementing universal VAS programs targeted at children from 6–59 months of age through semi-annual national campaigns, according to WHO recommendations [45]. However, our meta-analysis suggests that VAS has no consistent overall protective effect on the incidence or course of ARTI and that it increased the incidence of ARTI in the case of high-dose vitamin A supplementation, especially in children with a normal nutritional status. It is plausible that the studies included in this paper excluded children with signs of VAD, and VAS in excess of physiological requirements may cause an adverse effect on the immune system. Thus, we recommend that VAS for children aged 0 to 11 years should take the nutritional status and baseline serum retinol into consideration. Authorities should bear in mind that capsules should be distributed up to two times a year and at standard doses to avoid the accumulation of vitamin A toxicity.

Publication bias tends to favor studies with statistically significant results, rather than those showing no effect [57]. We believe our study was free of publication bias as most of the studies had negative results and the Egger test did not show any.

Our study has several limitations. We included studies that used different definitions of RTI and the small number of RCTs available for quantitative synthesis. Furthermore, we did not perform a subgroup analysis on age or gender, which may be helpful in explaining the great heterogeneity of results. Despite these limitations, our study is the first to include all ARTI RCTs and to explore the results by subgroup.

## 5. Conclusions

We conclude that VAS according to WHO recommendations has little value for children aged 0 to 11 years in the prevention of or recovery from acute respiratory infections, and it may cause a temporary immune dysregulation and lead to increased risks of infectious diseases when provided in excess of the WHO’s recommended dose. Thus, vitamin A should be provided based on the serum retinol levels and nutritional status of children, and a comprehensive nutritional approach should be taken.

## Figures and Tables

**Figure 1 nutrients-13-04251-f001:**
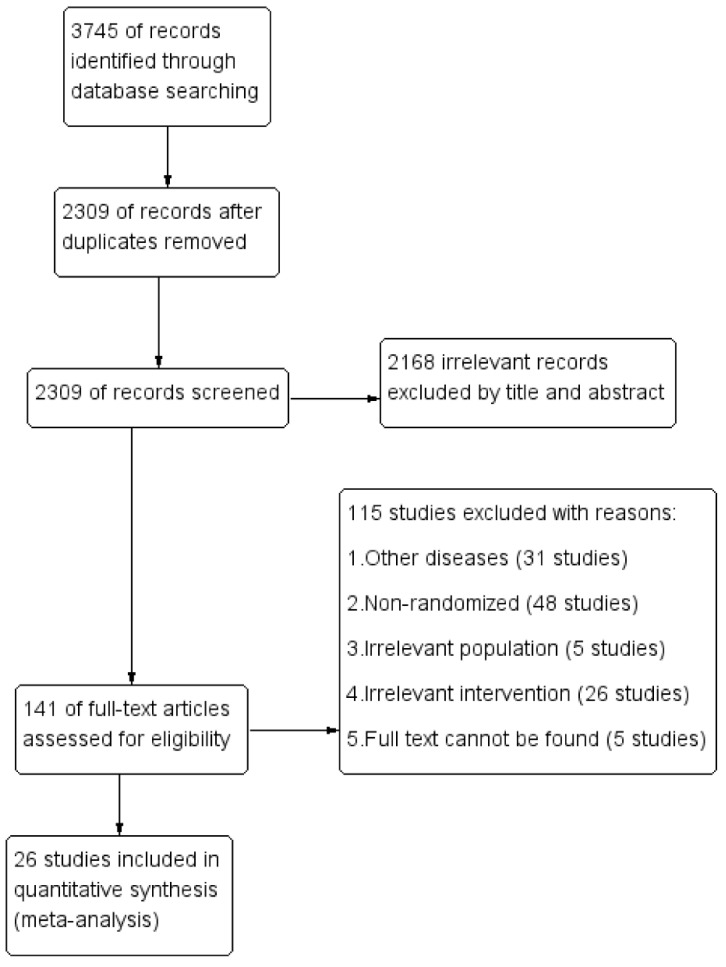
Prisma flowchart for the study selection process.

**Figure 2 nutrients-13-04251-f002:**
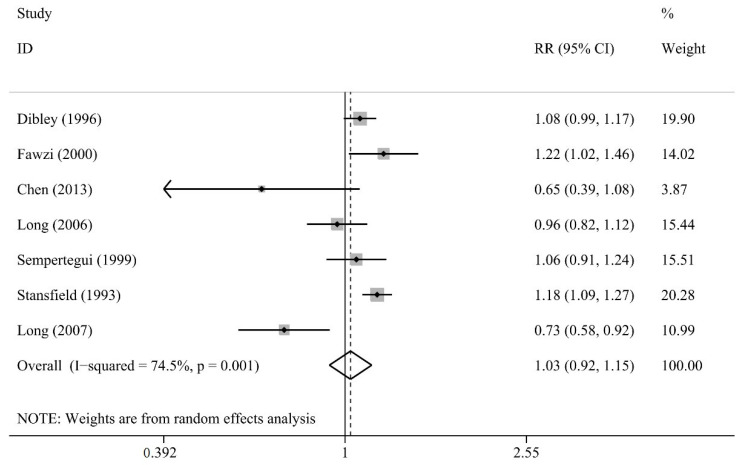
Comparison of Vitamin A versus Control, Outcome 1—incidence of acute respiratory tract infection (ARTI) at longest follow-up; RR: risk ratios or rate ratios.

**Figure 3 nutrients-13-04251-f003:**
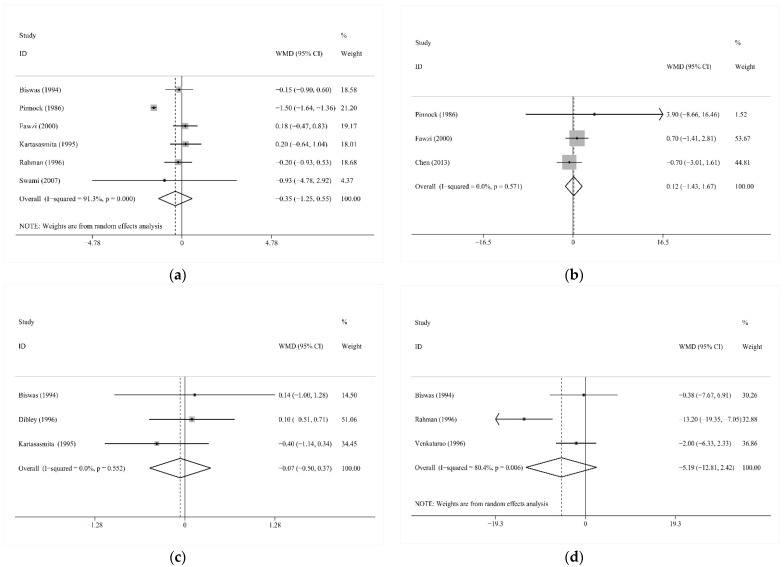
(**a**) Comparison of Vitamin A versus Control, Outcome 2—Mean number of episodes/child-year of acute respiratory tract infection (ARTI) at longest follow-up; (**b**) Comparison of Vitamin A versus Control, Outcome 3—Mean cough days of acute respiratory tract infection (ARTI) at longest follow-up; (**c**) Comparison of Vitamin A versus Control, Outcome 4—Mean duration of episodes due to acute respiratory tract infection (ARTI) at longest follow-up; (**d**) Comparison of Vitamin A versus Control, Outcome 5—Mean duration per child due to acute respiratory tract infection (ARTI) at longest follow-up; WMD: weighted mean difference.

**Figure 4 nutrients-13-04251-f004:**
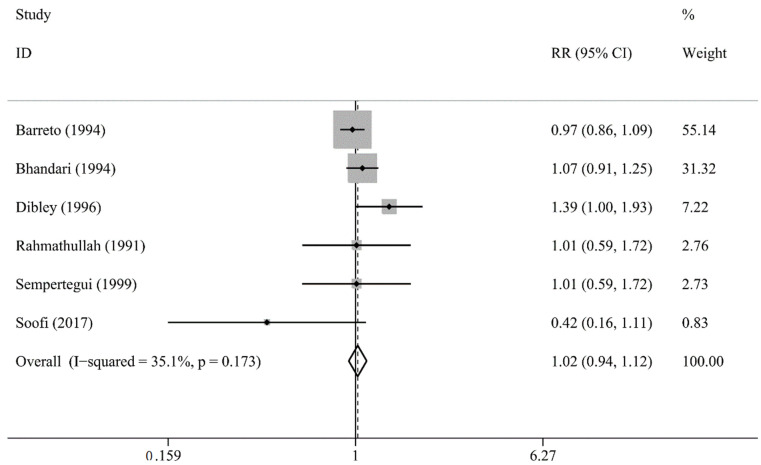
Comparison of Vitamin A versus Control, Outcome 6—Lower respiratory tract infection (LRTI) incidence at longest follow-up; RR: risk ratios or rate ratios.

**Figure 5 nutrients-13-04251-f005:**
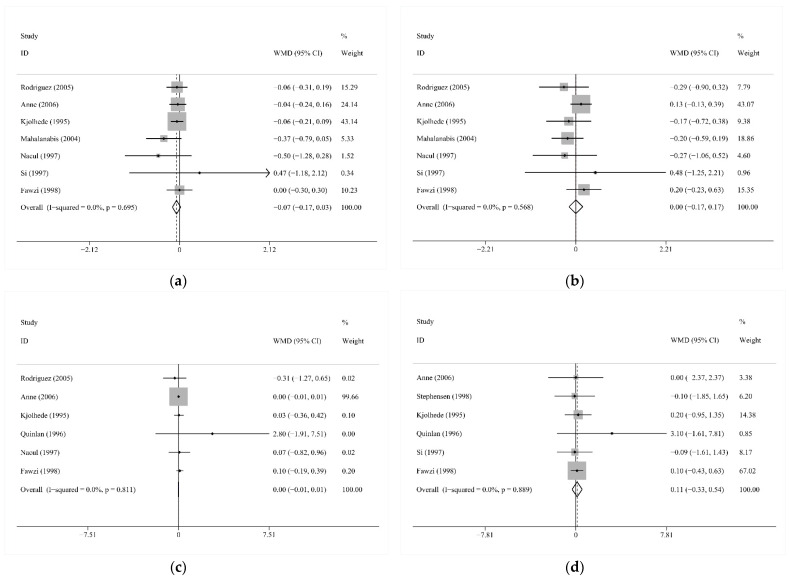
(**a**) Comparison of Vitamin A versus Control, Outcome 7—Days to resolution of fever due to lower respiratory tract infection (LRTI) at longest follow-up; (**b**) Comparison of Vitamin A versus Control, Outcome 8—Days for respiratory rate to settle due to lower respiratory tract infection (LRTI) at longest follow-up; (**c**) Comparison of Vitamin A versus Control, Outcome 9—Days to normalization of SpO2 due to lower respiratory tract infection (LRTI) at longest follow-up; (**d**) Comparison of Vitamin A versus Control, Outcome 10—Days in hospital due to lower respiratory tract infection (LRTI) at longest follow-up; WMD: weighted mean difference.

**Figure 6 nutrients-13-04251-f006:**
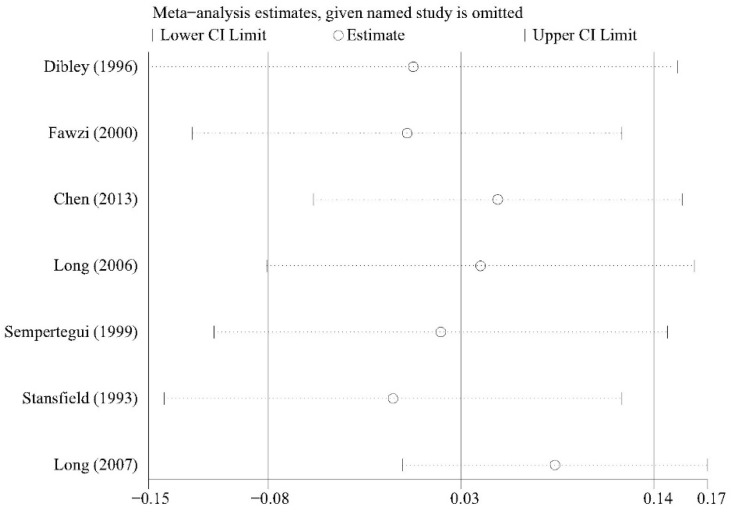
Sensitivity analysis of acute respiratory tract infection (ARTI) incidence.

**Figure 7 nutrients-13-04251-f007:**
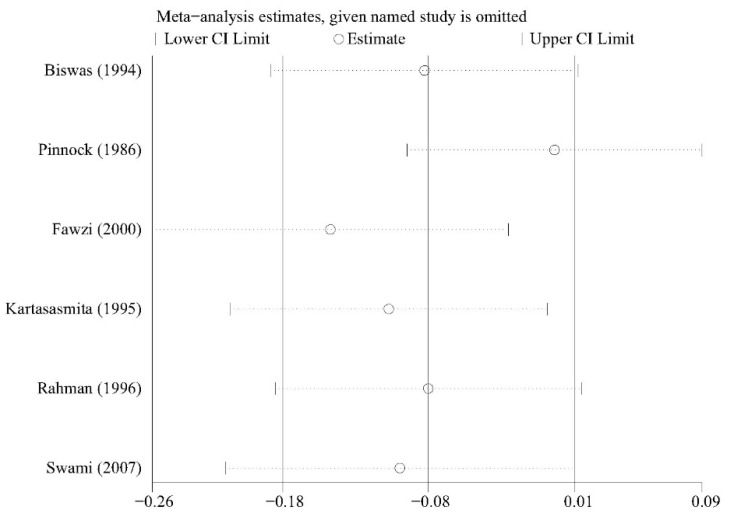
Sensitivity analysis of the mean number of episodes.

**Table 1 nutrients-13-04251-t001:** Characteristics of the included studies.

First Author/Location/Year	Age Range (Month)	Baseline Serum Retionl Status (μmol/L)		Intervention and Duration		Number per Limb		Outcome Measures	Overall Mortality of Study		
		IT	CT	<1 Year	>1 Year	IT	CT		Total	IT	CT
Anne/Australia/2006	<132	0.50	0.80	50,000 IU on Days 1 and 5	100,000 IU on Days 1 and 5	108	107	Days to normalization of SpO2; Days to resolution of fever; Days for respiratory rate to settle; Days in hospital	0	0	0
Kjolhede/Guatemala/1995	3–48	0.92	0.87	100,000 IU on admission	200,000 IU on admission	132	131	Days to normalization of SpO2; Days to resolution of fever; Days for respiratory rate to settle; Days in hospital	4	2	2
Mahalanabis/India/2004	2–24	0.71 ± 0.53	0.71 ± 0.62	33,333 IU twice daily for 4 d	33,333 IU twice daily for 4 d	38	38	Days to resolution of fever; Days for respiratory rate to settle	1	1	0
Quinlan/Chicago/1996	2–58	-		100,000 IU on admission	100,000 IU on admission	21	11	Days to normalization of SpO2; Days in hospital	0	0	0
Fawzi/Tanzanian/1998	6–60	-	-	200,000 IU over 2 d	400,000 IU over 2 d	346	341	Days to normalization of SpO2; Days to resolution of fever; Days for respiratory rate to settle; Days in hospital	21	13	8
Nacul/Brazil/1997	6–59	0.45 ± 0.34	0.38 ± 0.28	200,000 IU over 2 d	400,000 IU over 2 d	239	233	Days to normalization of SpO2; Days to resolution of fever; Days for respiratory rate to settle	4	2	2
Rodríguez/Ecuador/2005	2–59	1.26 ± 0.54	1.35 ± 0.59	50,000 IU on admission	100,000 IU on admission	121	118	Days to normalization of SpO2; Days to resolution of fever; Days for respiratory rate to settle	5	2	3
Si/Vietnam/1997	1–59	-	-	200,000 IU over 2 d	400,000 IU over 2 d	279	309	Days to resolution of fever; Days for respiratory rate to settle; Days in hospital	4	1	3
Stephensen/Peru/1998	3–120	0.24 ± 0.17	0.31 ± 0.24	100,000 IU on admission and 50,000 IU the nextday	200,000 IU onadmission and 100,000 IU the next day	48	47	Days in hospital	0	0	0
Bhandari/Govindpuri/1994	12–60	-	-	-	200,000 IU on admission	422	420	Incidence of Acute lower respiratory tract Infection	-	-	-
Biswas/Calcutta/1994	12–71	-	-	-	200,000 IU on admission	91	83	Mean number of episodes; Mean duration per episode; Mean duration per child	0	0	0
Rahmathullah/India/1991	6–60	-	-	8375 IU/week, 52 weeks	8375 IU/week, 52 weeks	7655	7764	Incidence of Acute lower respiratory tract Infection	-	-	-
Dibley/Indonesian/1996	6–47	-	-	103,000 IU/4 months, 2 years	206,000 IU/4 months, 2 years	396	386	Incidence of Acute lower respiratory tract Infection; Incidence of Acute respiratory tract infection	1	0	1
Chen/China/2013	36–72	1.15 ± 0.30	1.14 ± 0.27	-	200,000 IU on admission	95	104	Incidence of Acute respiratory tract infection; Cough (days)	-	-	-
Barreto/Brazil/1994	6–48	-	-	100,000 IU/4 months, 1 year	200,000 IU/4 months, 1 year	620	620	Incidence of Acute lower respiratory tract Infection; Cough (days)	4	2	2
Venkatarao/India/1996	0–12	-	-	200,000 IU at 6 months old	-	311	297	Mean duration per child	12	3	9
Long/Mexico/2006	6–15	-	-	20,000 IU/2 months, 1 year	45,000 IU/2 months, 1 year	180	183	Incidence of Acute lower respiratory tract Infection	-	-	-
Pinnock/Adelaide/1986	12–48	4.21 ± 0.15	4.08 ± 0.17	-	1500 IU/day, 5 months	76	71	Mean number of episodes; Cough (days)	-	-	-
Kartasasmita/India/1995	12–54	2.71 ± 0.65	1.60 ± 0.59	-	200,000 IU on admission and 6 months	126	143	Mean number of episodes; Mean duration per episode	-	-	-
Rahman/Bangladesh/1996	2.5	0.43 ± 0.24	0.42 ± 0.20	50,000 IU on 4 week, 8 week	-	84	81	Mean number of episodes; Mean duration per child	-	-	-
Sempertegui/Ecuador/1999	6–36	3.40 ± 0.93	3.49 ± 0.91	10,000 IU/week, 40 weeks	10,000 IU/week, 40 weeks	200	200	Incidence of Acute lower respiratory tract Infection; Incidence of Acute respiratory tract infection	-	-	-
Stansfield/Haiti/1993	6–83	-	-	100,000 IU/4 months, 1 year	200,000 IU/4 months, 1 year	8351	6993	Incidence of Acute respiratory tract infection	72	36	36
Fawzi/Tanzania/2000	6–60	-	-	100,000 IU/4 months, 1 year	200,000 IU/4 months, 1 year	289	285	Mean number of episodes; Cough (days); Incidence of Acute respiratory tract infection	-	-	-
Swami/Chandigarh/2007	12–60	-	-	-	200,000 IU on admission	276	252	Mean number of episodes	2	0	2
Long/Mexican/2007	6–15	-	-	20,000 IU/2 months, 1 year	45,000 IU/2 months, 1 year	97	98	Incidence of Acute respiratory tract infection	-	-	-
Soofi/Pakistan/2017	0–1	-	-	50,000 IU on admission	-	5380	5648	Incidence of Acute lower respiratory tract Infection	243	128	115

VA: vitamin A; IT: intervention group; CT: control group.

**Table 2 nutrients-13-04251-t002:** Quality of the studies included in the meta-analysis.

First Author	Random Sequence Generation	Allocation Concealment	Blinding of Participants and Personnel	Blinding of Outcome Assessment	Incomplete Outcome Data	Selective Reporting	Other Bias
Anne 2006	low	low	low	unclear	low	low	low
Kjolhede 1995	unclear	low	unclear	unclear	high	low	unclear
Mahalanabis 2004	low	low	low	low	low	low	low
Quinlan 1996	unclear	unclear	low	unclear	low	high	high
Fawzi 1998	unclear	low	low	unclear	low	low	low
Nacul 1997	low	low	low	low	low	low	low
Rodríguez 2005	low	low	low	low	low	low	low
Si 1997	unclear	unclear	low	unclear	unclear	unclear	high
Stephensen 1998	low	low	low	unclear	low	unclear	low
Bhandari 1994	low	low	low	low	low	low	low
Biswas 1994	low	low	low	low	high	unclear	unclear
Rahmathullah 1991	unclear	low	low	unclear	low	high	unclear
Dibley 1996	low	low	low	unclear	unclear	high	high
Chen 2013	unclear	high	low	unclear	low	unclear	high
Barreto 1994	low	low	low	low	high	low	low
Venkatarao 1996	unclear	low	high	high	high	unclear	high
Long 2006	low	low	low	unclear	high	low	unclear
Pinnock 1986	unclear	low	low	unclear	high	unclear	unclear
Kartasasmita 1995	low	unclear	unclear	unclear	low	unclear	high
Rahman 1996	unclear	low	low	unclear	high	low	high
Sempertegui 1999	low	low	low	unclear	high	low	high
Stansfield 1993	low	low	low	low	low	low	low
Fawzi 2000	unclear	low	low	unclear	low	low	unclear
Swami 2007	unclear	unclear	unclear	unclear	high	high	high
Long 2007	low	low	low	unclear	low	unclear	unclear
Soofi 2017	low	low	low	low	low	unclear	unclear

**Table 3 nutrients-13-04251-t003:** Subgroup analyses—incidence of acute respiratory tract infection.

	Subgroup	Number of Studies	RR	95%CI	*p*	Heterogeneity	
						I^2^	*p*
dose							
	high	4	1.131	(1.065, 1.200)	<0.001	19.3%	0.294
	standard	3	0.815	(0.643, 1.032)	0.090	59.4%	0.085
frequency							
	low	1	0.650	(0.392, 1.078)	0.095		
	medium	3	1.143	(1.066, 1.225)	<0.001	32.0%	0.230
	high	3	0.923	(0.764, 1.115)	0.406	70.3%	0.035
area							
	Asia	2	0.893	(0.552, 1.446)	0.645	73.5%	0.052
	Other areas	5	1.033	(0.893, 1.195)	0.658	79.2%	0.001
Nutritional status							
	stunted	3	0.821	(0.457, 1.474)	0.509	59.7%	0.083
	wasted	2	0.589	(0.208, 1.668)	0.319	52.3%	0.148
	normal	3	1.656	(1.302, 2.106)	<0.001	14.9%	0.309

RR: rate ratios (events per child-year) or RRs (events per child); dose: standard (up to 100,000 IU for children aged 0 to 11 months and 200,000 IU for children aged 12 months to 11 years every 4–6 months); high (greater than standard); frequency: low (one dose 4-plus-month); medium (one dose every 4 months); high (doses more than once in 4 months); CI: confidence intervals.

**Table 4 nutrients-13-04251-t004:** Subgroup analyses—mean number of episodes.

	Subgroup	Number of Studies	WMD	95%CI	*p*	Heterogeneity	
						I^2^	*p*
dose							
	standard	4	−0.683	(−1.684, 0.318)	0.181	86.9%	<0.001
	high	2	0.188	(−0.327, 0.702)	0.475	0.0%	0.971
frequency							
	low	4	−0.081	(−0.522, 0.360)	0.719	0.0%	0.860
	median	1	0.180	(−0.473, 0.833)	0.589		
	high	1	−1.500	(−1.643, 1.357)	<0.001		
area							
	Asia	4	−0.081	(−0.522, 0.360)	0.719	0.0%	0.860
	Other areas	2	−0.691	(−2.337, 0.954)	0.410	95.9%	<0.001

WMD: weighted mean difference; dose: standard (up to 100,000 IU for children aged 0 to 11 months and 200,000 IU for children aged 12 months to 11 years every 4–6 months); high (greater than standard); frequency: low (one dose 4-plus-month); medium (one dose every 4 months); high (doses more than once in 4 months); CI: confidence intervals.

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
