# Peer review of "Excessive Vitamin A Supplementation Increased the Incidence of Acute Respiratory Tract Infections: A Systematic Review and Meta-Analysis"

_nutrients, 2021, doi:10.3390/nu13124251_

Round 1

Reviewer 1 Report

The article by Zhang and colleagues, provides an intriguing review and meta-analysis of the incidence of ARTI and excess of Vitamin A supplementation.

The Authors are advised to:

  • provide more up-to date references for Lines 31-34 of the article
  • typo check on line 196, use capital E for "egger test"
  • Use a consistent style for Tables and Figures captions (check for character dimensions)
  • Provide a high resolution, readable image for Figure 3a, 3b, 3c, and 3d, 4a, 4b, 4c, and 4d.
  • Provide a readable, high resolution for Fiugres 6 and 7 (a white background would be preferable).

Reviewer 2 Report

very nice review article, would recommend adding more references from Patrick Borel, Johannes von Lintig, and Glenn Lobo groups in the introduction and discussion, specifically related to Vitamin A intake, absorption, storage, and delivery.

 Comments:

Abstract: not clear what is meant by "children living in settings"..this needs to be re-written.

Fig. 3 and 4: Font sizes are too small to read, reconsider increasing the sizes.

Round 2

Reviewer 2 Report

Authors have made corrections and added relevant references to their manuscript.